# Review and Improvement of Chemical Hazard Risk Management of Korean Occupational Safety and Health Agency

**DOI:** 10.3390/ijerph18179395

**Published:** 2021-09-06

**Authors:** Saemi Shin, Sang-Hoon Byeon

**Affiliations:** School of Health and Environmental Science, Korea University, Seoul 02841, Korea; saemishin@naver.com

**Keywords:** CHARM, COSHH Essentials, risk assessment, control banding, permissible exposure limit

## Abstract

In 2012, the Korean Occupational Safety and Health Agency developed Chemical Hazard Risk Management (CHARM) as a risk assessment tool. This study aims to reorganize the CHARM technique by complementing its logical loopholes, while evaluating the risk to enterprises and verifying this technique by applying it to some enterprises in Korea. The optimized technique changed the method of quantitative assessment and evaluation criteria, matched the risk level with the required control level, and specified the use of control practice. For the target enterprises, for several assessment methods, risk levels, hazard bands, exposure bands, and the risk assessment results were derived, and the same types of options were compared. Fewer informational methods resulted in more conservative results of risk levels and hazard bands. Since the control status of the enterprises could not be confirmed and the substances handled at the target enterprises were limited in this study, a follow-up study should be performed with more target materials and additional information on the current control status of the enterprises.

## 1. Introduction

Chemical substances are widely used; however, when mishandled, they can cause environmental hazards and various occupational diseases [1]. Recently, in Korea, chemical accidents (such as the hydrofluoric acid leakages in Gumi in 2012, Cheongju industrial complex in 2013, and Samsung Electronics at Hwaseong in 2013 [2]) and their lasting effects [3] have become an issue; thus, chemical control systems were renovated in 2013 [4]. Carcinogens have also become a social concern owing to instances of lung cancer in workers exposed to asbestos as well as leukemia in workers employed at semiconductor factories [5]; consequently, the adverse effects of chemicals are a major concern in Korea. Considering that there are 45,000 kinds of chemical substances distributed annually in Korea, which accounts for approximately 3% of the global market [6], chemical control is extremely important and highly necessitated.

According to the 2019 Korean Labor Force Survey at Establishments, there are 18,881,701 workers in all businesses in Korea, of which 15,957,656 are regular workers [7]. According to the Occupational Safety and Health Act, the Korean Ministry of Employment and Labor classifies 706 hazardous chemicals into 7 groups, conducts a Work Environment Survey every 5 years, and manages exposure in the workplace. According to the 2019 Korean Work Environment Survey, the estimated number of workplaces and the estimated average number of workers by the sample survey for each chemical substance group are derived, and the total number of workers handling hazardous chemicals is estimated to be 6,773,610 [8].

The framework of the Korean chemical control system, which is used to prevent adverse effects from chemical substances, manages chemical substances by listing them based on their occupational exposure limit (OEL). According to the Occupational Safety and Health Act enacted in 1982, substances used in the workplace are classified into harmful agents for which occupational exposure limits are required to be set, materials the manufacture, etc., of which is prohibited, materials subject to permission, harmful factors to be maintained at or below the permissible level, harmful substances to be controlled, harmful substances to be specially controlled, etc., and the control level of a substance is determined according to recommendations provided by the law for that substance [9].

The risk level of each material is assessed on a national basis; therefore, there is a possibility that the control level does not match the risk level at an individual workplace. In addition, the process of setting OELs experiences limitations because analysis methods used for obtaining toxicological data of chemical substances are slow and cannot keep up with the rate of increase of chemical substances [10]. Furthermore, it is difficult for the industry or government to impose work standards through administrative measures, such as guidelines or self-regulation, in the private sector owing to the technical complexity of setting OELs, lack of adequate databases and experts, and enormous amount of substances [11].

To support the management of chemical substances in small and medium-sized enterprises, the UK Health and Safety Executive developed the Control of Substances Hazardous to Health Essentials (COSHH Essentials), a control banding technique that determines the management method by assigning the qualitative work environment characteristics of the enterprises to a hazard and exposure prediction band [12]. In 1998, they published a series of papers on control banding strategies [13,14,15]. Qualitative tools were used for assessing the risk of these chemicals, creating solutions, and implementing control measures in the fields of radiation, lasers, and biosafety in the 1980s. These tools were subsequently applied to the pharmaceutical industry in the 1990s [16,17]. COSHH Essentials has since been established as a versatile and verified control banding technique [14,15,18,19].

In 2012, the Korean Occupational Safety and Health Agency (KOSHA) developed and distributed Chemical Hazard Risk Management (CHARM). This is a risk assessment tool based on the COSHH Essentials banding method, in which individual enterprises self-evaluate and derive appropriate management methods [18,19]. CHARM involves a quantitative risk assessment that is based on monitoring data collected through the work environment monitoring system, and it also performs qualitative risk assessment using the hazard and exposure bands (H and E bands) that are based on the hazard statements of the Global Harmonized System of classification and labeling of chemicals (GHS) and the amount and physical properties of the substance, respectively. If no work environment monitoring data can be derived but an OEL exists, the qualitative risk assessment process is followed; however, the OEL is directly compared with the target airborne concentration [13] of the hazard statement to derive the H band. If quantitative values are available, such as work environment monitoring results or OELs, the evaluation is performed using this information and subsequently applied as a result. CHARM has been verified through a comparison of risk assessment results of the CHARM and COSHH Essential models [20], along with a comparison of the qualitative and quantitative evaluation results of CHARM [21].

Tischer [18] applied three aspects of model evaluation to the COSHH Essentials model. These aspects that were used for validating the models include the internal (conceptual) validation of model assumptions and structures, external (performance) validation of model predictions corresponding to specialized industrial hygiene monitoring data, and operational analysis of the understanding and implementation of the model results of the target group. Most publications on model validation are interested in mathematical models designed for different fields of study [22]. As no reports have conceptually verified Chemical Hazard Risk Management (CHARM), it is necessary to evaluate the conceptual validity of CHARM.

One problem with CHARM is that the concepts of hazard or exposure used in quantitative and qualitative risk assessments are not the same. In quantitative risk assessment, risk is evaluated using the hazard band (H band) derived from occupational exposure limit (OEL) and the exposure band (E band) derived from the ratio of exposure concentration and OEL. In the qualitative risk assessment, such as COSHH Essentials, risk is evaluated using the H band derived from the hazard statement, to which the target airborne concentration is assigned, together with the E band derived from the combination of the amount and physical properties assigned to the exposure-predicted concentration. The H band used in quantitative and qualitative risk assessments is essentially the same concept, but the E band is not. The concept of E band in quantitative risk assessment is based on exposure concentration versus exposure limit; however, it is considered a risk in the qualitative risk assessments of CHARM and COSHH Essentials.

In addition, despite using the same H and E bands as COSHH Essentials in qualitative risk assessment, the method of deriving the risk or control level is different. The approach of COSHH Essentials should be critically applied according to the country’s situation [23]. However, it may be inappropriate to arbitrarily change a method that has been developed based on considerable evidence and verified without an appropriate basis, rather than using it as it is. The International Labor Organization Chemical Control Toolkit (ILO Toolkit), which has been produced in collaboration with the UK Health and Safety Executive and the International Occupational Hygiene Association, exhibits a different hazard classification than that of COSHH Essentials. Jones and Nicas [19] did not positively assess that the change in control banding strategy would undermine users’ confidence; therefore, the authors of the ILO Toolkit should rethink their risk classification plans.

Finally, in CHARM, risk assessment is conducted to inform whether the current control status is acceptable or not according to the degree of risk. Subsequently, from the list of work environment control methods, it selects and applies the method applicable at the workplace through the employers; finally, it reassess the risk through feedback. Each control method in the list is simply listed without setting its relative efficacy; consequently, the importance of control banding in guiding users how to manage chemical control [24,25] is not being addressed. Therefore, it is necessary to further refine the method to derive the control approach.

This study aims to reorganize the technique by complementing the logical loopholes of CHARM, evaluate the risk to enterprises, and verify the proposed technique by applying it to some enterprises in Korea.

## 2. Materials and Methods

### 2.1. Development of the Method

The overall flow of the method is illustrated in Figure 1. The method of calculating the risk level varies depending on the presence or absence of data on work environment monitoring and OELs. Substances are designated as harmful agents subject to work environment monitoring, and harmful agents for which occupational exposure limits are required to be set, according to Article 105 of the Occupational Safety and Health Act, are deemed harmful. Since all harmful agents subject to work environment monitoring have OELs, three possibilities exist: both the work environment monitoring result and OELs are available, only OELs are available, and no data are available. The risk level is used to determine the engineering control approach of the workplace, which in turn is used to determine the overall control approach of the workplace by combining it with other administrative controls, such as education, protective equipment, and cleaning.

If results of the work environment monitoring are available along with occupational exposure limits (OELs), the risk level is calculated according to Table 1. If the work environment monitoring results are not available, a hazard band (Table 2) that assigns an OEL (if available), a hazard statement that assigns a category of target airborne concentration (if OEL data is not available), and an exposure band (Table 3) that assigns a category of exposure are used to derive the amount and properties (dustiness or volatility) of the substance.

When calculating the risk level from the hazard and exposure bands, the values of each band are converted into a score according to Table 4; subsequently, the exposure score is corrected according to Table 5, and the values are added to calculate the risk score. At this time, the definition of working hours in Table 5 has been provided according to the Korean local rules on occupation safety and health standards. The term “temporary work” implies that a person works less than once a month and less than 10 h per month, or for 10–24 h every month. “Short-term work” implies that a person works less than once a day and less than 1 h per day. According to Table 6, the risk level was determined using the risk score.

By combining the risk level and current control status, the engineering control approach was determined as shown in Table 7. The overall control approach is determined by combining the engineering control approach and the legal mandates or recommendations for substances, as shown in Table 8 and Figure 2.

### 2.2. Validation of the Method

To validate the method proposed in this study, the risk associated with some processes was evaluated using the work environment monitoring data of the harmful substances that need to be maintained at or below the permissible levels of 2014.

Harmful substances that need to be maintained at or below permissible levels are those substances for which companies are obligated to comply with the permissible exposure level and reinforced regulations (compared to the occupational exposure limits, which exist only as recommended standards). All harmful substances whose concentrations need to be maintained at or below permissible levels have been observed in work environment monitoring. Table 9 shows the list of substances that are designated as harmful and should be maintained at or below permissible levels of 2014.

According to the Occupational Safety and Health Act, substances were measured and monitored in a working environment, and the resulting data were submitted to the KOSHA. All chemical substances with an annual circulation of 1 ton or more were reported to be manufactured, used, stored, and disposed of through a statistical survey on chemicals in accordance with the Chemical Control Act, and the Ministry of Environment is obligated to disclose them to the public. For enterprises handling harmful factors to be maintained at or below permissible levels in a single process, 1164 workplaces that matched the results of the publication on the statistical survey on chemicals were selected using the names and addresses of the enterprises as matching keys.

For target workplaces and processes, the three risk levels that can be derived by this method were derived along with the hazard band (H band), that constitutes two types of qualitatively derived hazards, the exposure band (E band), and the risk assessment results of CHARM and COSHH Essentials.

For substances that were not detected, measured values were replaced with 1/2 of the detection limit. Some values remain unknown because the detection limit for each institution has not been collected nationally; however, the detection limits derived from the guidelines provided by KOSHA for analyzing harmful agents subject to work environment monitoring were cited, and values less than 1/2 of the detection limit were replaced with 1/2 of the detection limit.

In the national statistical data used in this study, the internal control status of the enterprises was not provided. For the step of correcting the exposure score using ventilation conditions by employing the proposed technique and CHARM, the internal control status was collectively assumed to be general ventilation. Since work environment monitoring is performed for processes that are not temporary or short-term, the exposure score correction according to working time was not performed for all data. The volatility of liquid substances was determined by assuming the process temperature to be room temperature. Meanwhile, in the case of solid substances, work environment monitoring was only carried out in a state where breathing could provide exposure to the substances; therefore, all of the workplaces were assumed to have the highest dust concentrations.

The relationship between quantitative risk and qualitative risk was derived by testing the difference in the ratio of the three risk levels that can be derived from this method for the target data. By testing the difference in the ratio of the two H bands for the target data, the relationship between the actual occupational exposure limit and the target airborne concentration that assigned the hazard statement was identified.

Multiple comparisons were performed to determine if there was a difference in the actual measured values between the E bands derived by this method for the target data. The mean difference was tested to determine whether there was a difference between the actual measured value and the predicted concentration assigned to the E band.

The relationship between each value was derived from each control banding technique by testing the difference in the ratio of the values (the risk level of this technique, the risk level of CHARM, and the control approach of COSHH Essentials) for the target data to indicate the level of risk.

## 3. Results

We specifically promoted the following innovations compared to CHARM. The standard of quantitative risk assessment was changed to ratio of exposure concentration to exposure limit, and exposure was changed to the average of exposure assuming a constant geometric standard deviation from the standard of maximum exposure. The interval for each evaluation grade was adjusted in such a way that a mechanistic model was applied consistently rather than an arbitrary criterion. A separate risk score and risk level system were established from CHARM, and the risk level and management level recommendations were matched. An algorithm was established that utilizes the checklist that was presented as a whole without any recommendations from CHARM, and legal obligations, control approaches, and alternative controls that can be referred to at the workplace are assigned for each control practice.

For target enterprises, the risk level derived from the ratio of work environment monitoring results and occupational exposure limit (OEL) (hereinafter, “quantitative”) was level 1 at 915 (78.6%), level 2 at 115 (9.9%), level 3 at 123 (10.6%), and level 4 at 11 (0.9%) locations among 1164 locations for all materials, level 1 at 583 (82.3%), level 2 at 54 (7.6%), level 3 at 67 (9.5%), and level 4 at 4 (0.6%) locations among 708 locations for handling solid materials, and level 1 at 332 (72.8%), level 2 at 61 (13.4%), level 3 at 56 (12.3%), and level 4 at 7 locations (1.5%) among 456 locations for handling liquid materials.

The risk level derived from the hazard band (H band) based on OEL and the exposure band (E band) based on usage and physical properties (hereinafter, “qualitative 1”) was level 1 at 149 (4.2%), level 2 at 212 (18.2%), level 3 at 338 (29.0%), and level 4 at 565 (48.5%) locations among 1164 locations for all materials, level 2 at 26 (3.7%), level 3 at 273 (38.6%), and level 4 at 409 (57.8%) locations among 708 locations for handling solid materials, and level 1 at 49 (10.7%), level 2 at 186 (40.8%), level 3 at 65 (14.3%), and level 4 at 156 (34.2%) locations among 456 locations for handling liquid materials.

The risk level derived from the H band based on hazard statements and the E band based on usage and physical properties (hereinafter, “qualitative 2”) was level 4 at 1164 locations (100%) for all target materials. Figure 3 shows the summation of the distribution of the frequency of risk level for each phase, quarter, and level. To determine whether there were differences in the three risk levels derived for one target enterprise, the Friedman test, which checks the difference in the sum of rankings between the corresponding samples, was conducted, and it was confirmed that *p* < 0.001 for all phases of solid and liquid and that the difference was significant.

The H band derived from OEL was categorized as band 1 at 176 (15.7%), band 2 at 122 (10.4%), band 3 at 819 (70.4%), and band 5 at 47 (4.0%) locations among 1164 locations for all materials, band 2 at 63 (8.9%) and band 3 at 645 (91.1%) locations among 708 locations for handling solid materials, and band 1 at 176 (38.6%), band 2 at 59 (12.9%), band 3 at 174 (38.2%), and band 5 at 47 (10.3%) locations among 456 locations for handling liquid materials. The H band derived from the hazard statements was band 5 at 1164 locations (100%) for all target materials. Figure 4 shows the summation of the distribution of the frequency of the H band for each phase, quarter, and level. The Wilcoxon signed-rank test, which checks the difference in the sum of ranks between the corresponding samples, was conducted to check whether the difference between the two H bands was derived for each target process. It was observed that *p* < 0.001 for both the solid and liquid phases, indicating that the difference was significant.

The E band derived by usage and physical properties was categorized as band 1 at 18 (1.5%), band 2 at 459 (39.4%), band 3 at 640 (55.0%), and band 4 at 47 (4.0%) locations among 1164 locations for all materials, band 2 at 266 (37.6%), band 3 at 418 (59.0%), and band 4 at 24 (3.4%) locations among 708 locations for handling solid materials, and band 1 at 18 (3.9%), band 2 at 193 (42.3%), band 3 at 222 (48.7%), and band 4 at 23 (5.0%) locations among 456 locations for handling liquid materials.

The geometric mean (GM) and geometric standard deviation (GSD) of the measured values for each phase and band were GM = 1.46 × 10^−3^ and GSD = 3.53 at band 2, GM = 1.58 × 10^−3^ and GSD = 3.08 at band 3, and GM = 2.05 × 10^−3^ and GSD = 7.07 at band 4 for solid materials, and GM = 9.88 × 10^−4^ and GSD = 233.17 at band 1, GM = 1.09 × 10^−2^ and GSD = 55.63 at band 2, GM = 8.05 × 10^−2^ and GSD = 27.22 at band 3, and GM = 7.81 × 10^−3^ and GSD = 4.47 at band 4 for liquid materials.

Before performing the test to assess the difference in the mean value between bands or between each band and the actual exposure, a normality test was first performed on the exposure data.

By testing the normality of the log value of the monitoring results for all data in each phase using the Shapiro–Wilk test and subsequently checking visually through qqplot, the *p*-value was observed to be less than 0.001 in both phases. The shape of the qqplot is shown in Figure 5, confirming that there was no normality in the data.

By testing whether there is a difference in the actual measured values between bands for each phase using the Dwass–Steel–Critchlow–Fligner (DSCF) test (a nonparametric multiple comparison method), in the case of the solid phase, there was a significant difference between bands 2 and 4 (*p* = 0.035) and no significant difference between bands 2 and 3 (*p* = 0.894) and bands 3 and 4 (*p* = 0.096; *p* = 0.849). In the liquid phase, band 3 was found to have a significantly different value from those of the other bands (*p* = 0.024 for band 1, *p* < 0.001 for band 2, *p* = 0.031 for band 4), while there was no significant difference between the remaining bands (bands 1 and 2 *p* = 0.542, bands 1 and 4 *p* = 0.998, bands 2 and 4 *p* = 0.511). Figure 6 shows the overall distribution of exposures for each phase and band.

The random number was generated and logged, where the lower 5% represented the lower limit of the band quota and the top 5% signified the upper limit of the band quota. The difference between the measured value and the log value was tested according to the Mann–Whitney test (a nonparametric mean difference test for the data). It was confirmed that *p* < 0.001 in all phases and bands, and the measured value was found to be different from the predicted exposure. Figure 7 shows the data distribution pattern for each phase and band, which confirmed that the measured value was lower than the predicted exposure.

For the target workplace, the risk level derived from this technique is quantitative because work environment monitoring results are available; furthermore, the results of CHARM are the same. The risk level derived by CHARM was level 2 at 949 (81.5%), level 3 at 196 (16.8%), and level 4 at 19 (1.6%) locations among 1164 locations for all materials, level 2 at 604 (85.3%), level 3 at 95 (13.4%), and level 4 at 9 (1.3%) locations among 708 locations for handling solid materials, and level 2 at 345 (75.7%), level 3 at 101 (22.1%), and level 4 at 10 (2.2%) locations among 456 locations for handling liquid materials. The control approach derived by the COSHH Essentials model was level 4 at 1164 locations (100%) for all target materials. Figure 8 shows the summation of the frequency distribution for each method and the results. The Friedman test was performed to determine whether the results of the three risk assessments derived for each target process were different, and it was determined that *p* < 0.001 for all phases, indicating that the difference was significant.

## 4. Discussion

### 4.1. The Overall Structure of This Method

This study aimed to reorganize the CHARM technique by complementing its logical loopholes. The logic used for quantitative and qualitative risk assessment in a dualized risk assessment system is different from that used for quantitative risk assessment. The ratio of exposure concentration and occupational exposure limits (OELs) is assigned to the exposure band (E band); for risk assessment, the predicted exposure concentration is also assigned to the E band.

KOSHA’s publication [26] that describes CHARM only cites COSHH Essentials as a referenced control banding technique; however, the overall risk assessment structure of CHARM is more similar to the American Industrial Hygiene Association (AIHA) prioritization [27] based on OELs. The method of calculating a risk score of 1 to 16 by classifying the hazard and exposure bands as 1–4 and multiplying them with each other is similar to the AIHA method, which prioritizes the calculation of the health risk rating (1 to 16) by multiplying the health effect rating and exposure rating. The value used to calculate the AIHA exposure rating is also the exposure concentration to OEL ratio.

The concept of risk used in AIHA is interpreted as the product of probability and loss, which is similar to that defined in the engineering field [28,29]. Accordingly, the exposure rating is derived from the probability of the occurrence of an adverse effect, and the health effect rating is derived from the reversibility of the adverse effect [27]. However, in COSHH Essentials, risk is interpreted as the ratio of exposure (concentration) and hazard (toxicity) according to a typical risk assessment structure that is commonly used in the health field [30].

To reorganize CHARM, it was necessary to select or integrate one of the two different schemes that are built into the CHARM model. In this study, reorganization was performed according to the scheme of COSHH Essentials, owing to the following reasons: First, it was determined that the risk assessment logic used in the health field was more suitable for assessing the health risks of chemical substances. Second, COSHH Essentials, which links the combination of hazard and exposure to a control approach, was judged to be more suitable than the method of the American Industrial Hygiene Association for prioritizing purposes. This is because it does not give control to the calculated health risk rating by allowing the enterprises to manage and reduce the risk of the workplace by themselves.

### 4.2. Development of Quantitative Evaluation Criteria

For qualitative risk assessment, the entire process of COSHH Essentials was used as it is. In the case of quantitative risk assessment, hazard and exposure are not necessarily banded separately, and the ratio between the exposure concentration and the occupational exposure limit (OEL) was categorized and assigned to the risk level. Since the control banding tool is not intended to provide exclusive exposure predictions and hazard grouping models [25,31], hazard and exposure ratings were not provided separately. The risk level used in this method indicates how high a control level should be compared to the present value, and an engineering control approach is derived by combining the risk level and the current control level.

In Korea, the determination of the exposure level by law is based on whether the maximum value exceeds the OEL. Similarly, in CHARM, the category of risk level is determined by comparing the maximum value of exposure with the OEL. However, the level of exposure to the workplace can vary depending on the time and place. Statistical methods to reflect variability in exposure assessment have been developed since the 1960s [32,33,34] and subsequently fused with the guidelines of each institution for implementing industrial hygiene practices in various countries. Estimation of the percentage of days expected to exceed the estimated OEL and its use as a variable for exposure assessment is recommended by the Institut National de Recherche et de sécurité in France, British Occupational Hygiene Society/Nerlandse Vereniging voor Arbeidshygiëne, and Comite Europeen de Normalisation, and this method is implemented by extracting a random sample from the exposure distribution, and forms the basis of French regulations [35].

However, despite the well-established theoretical structure, there are only a few lognormal statistical tools that practitioners tend to use [36], and it is almost impossible for non-professional enterprises to use them. The National Institute for Occupational Safety and Health determines the action level, which represents the criterion that determines the necessity of workplace management based on the daily mean concentrations. The action level is a criterion for determining the probability that at least 5% of the true daily average exceeds the random upper tolerance limit to be less than 5%, and it includes a probabilistic concept. However, the National Institute for Occupational Safety and Health does not require calculation of this value for individual enterprises; instead, it uses the median geometric standard deviation (GSD), which is derived by analyzing the results of 55 individual workplace measurements as a representative value of GSD [37]. Similar to this method, when a high GSD of >2 was assumed from a conservative point of view, the arithmetic mean data were presented as a criterion for suggesting the necessity of control. These data were obtained by performing work environment monitoring for days where the 95% upper confidence limit of the 95th percentile exceeded the OEL (5% of the OEL).

In the case of COSHH Essentials, the relative efficacy of the control approach was assumed to be 10 for local exhaust ventilation and 100 for the containment; however, in this study, it was judged that a more rigorous evaluation of the effect of local control was necessary in order to apply a factor to the quantitative evaluation. One promising approach to help understand the inhalation exposure process is to use a source-receptor model, while providing approximate exposure through a deterministic exposure modifier [38]. A mechanistic model based on this approach was developed by Cherrie et al. [39] and validated against workplace measurements [39,40].

Cherrie and Schneider used local control as a modifying factor for exposure and recommended using 1 for no local ventilation, 0.3 for some forms of local control, and 0.1 for well-designed and maintained local control. In this study, the relative efficacies of 3 for local exhaust ventilation and 10 for containment were observed. Accordingly, the standard of the exposure band (E band) was set to 0, 0.05, 0.15, and 0.5, for the calculating the arithmetic mean data of the work environment monitoring compared to the occupational exposure limit (OEL).

### 4.3. Development of Qualtitative Evaluation Criteria

The qualitative evaluation proceeds in a manner similar to COSHH Essentials, which links the band to the target air concentration range. However, there are differences in several areas. First, in terms of the composition of the hazard band (H band), the value for the substance group (group E) in COSHH Essentials, to which an exposure standard of “0” has been allocated, was reset. Substances with OEL and non-OEL data were evaluated in one band because there were no substances with zero OEL. In the case of CHARM, the H band was set to 4 and group E was virtually deleted; however, in this study, it was determined that band 5 should be installed to prevent the group corresponding to group E from being underestimated when compared to COSHH Essentials, which is essential for qualitative evaluation. The allocation value was allocated to 1/10 of band 4 by referring to the recommendation that COSHH Essentials should alternatively control the Assigned Protection Factor 2000 level for group E substances.

Second, in the composition of the E band, by reflecting the characteristics of the Korean chemical control system where the control criteria for the substance may have already been determined, the method was constructed to ensure that the exposure determined by the amount and the physical properties of the substance can be corrected according to the current control condition. In CHARM, the current ventilation condition is reflected in the exposure, and in this method, the exposure score can be corrected through the existing ventilation condition. In addition, exposure correction was applied according to the working hours included in COSHH Essentials, and not those included in CHARM. Since it is not easy to determine the working hours or frequency at the enterprises, the concepts of ‘temporary work’ and ‘short-term work’ that determine whether or not a substance is to be monitored were used according to Korean regulations.

Third, depending on the characteristics of the Korean chemical control system, wherein the legal control standard for substances may have already been determined, the risk level to which the necessary relative efficacy is assigned at the current control status has been derived separately from the control approach calculated by combining the current control status.

Finally, the control approach is divided into three categories: legally mandated or recommended control, engineering control required according to the risk, or alternative control, which is applicable when engineering control is not possible according to the legal classification of the substance or the risk of the process.

### 4.4. Application of the Method and Analysis of Results

By using the developed method with the data of the target enterprises, it was confirmed that risk level 1 for all phases (i.e., the recommendation to maintain the control level) occupied the highest percentage in the quantitative risk assessment results. Existing studies on workplace environments in Korea also showed that the rate of excess concentration determined via work environment monitoring, the rate compared to the OEL, and the detection rate of carcinogens were all low [41,42,43,44,45].

Conversely, in the qualitative risk assessment, the results of qualitative level 1 showed that 100% of solids (708 of 708) and 90.3% of liquids (407 of 456) were required to increase the management level to risk level 2 or higher. The results of qualitative level 2 showed that risk level 4 was 100% for both solids and liquids. There was a significant difference between the quantitative risk assessment, the qualitative risk assessment with the occupational exposure limits assigned to the hazard band (H band), and the qualitative assessment with the hazard statement assigned. To prevent false negatives (which the management level determined based on the estimated value) from reaching the required management level when determined based on the actual exposure, it is reasonable to show conservative results by employing an evaluation method that uses less information. The lower the amount of information, the higher the risk level; however, the risk level derived from quantitative risk assessment is higher than that derived from qualitative 1, with one case (0.14%) for solids and 46 examples (10.1%) for liquids.

As can be seen from the analysis results of this study, the exposure estimate derived by the method of COSHH Essentials is significantly higher than the actual exposure. Since COSHH Essentials itself is designed with a conservative technique [15,25,46,47], qualitative results are more likely to be more conservative than the quantitative results. However, considering the criteria for evaluating risk, the quantitative assessment of this technique is more conservative than the qualitative assessment, and the difference in actual exposure by the exposure band (E band) is not clear; therefore, the difference between the estimated exposure and the actual exposure in the case of a low-grade E band is less than that in its high-grade counterpart. These results negatively affect the results of the qualitative evaluation when compared with the results of the quantitative evaluation. The fact that the actual exposure amount is not significantly different for each E band implies that there is a limit to the qualitative exposure evaluation itself. The failure to reflect the fluctuations in the exposure amount (according to management conditions such as ventilation) in the calculation of the exposure score, along with the use of only exposure data for limited substances, may reduce the accuracy of qualitative exposure prediction. In the future, it will be necessary to verify the exposure estimation results of COSHH Essentials applied by CHARM for various exposure data in Korea.

When comparing the two types of H bands calculated by this technique, it can be observed that band 5 is derived for all H bands using the hazard statement, and this is unlike the H bands based on the exposure criteria derived using various bands. Harmful substances to be maintained at or below the permissible level require high-level control; that is, compliance with the exposure limit is legally enforced. All of these substances exhibit irreversible hazards; therefore, when the qualitative hazard assessment criteria used in COSHH Essentials were applied, all were classified as the highest band. The verification of the method developed in this study is only for enterprises that handle highly toxic substances, and this may cause bias. Therefore, it is necessary to expand and verify the target if more data are secured in the future.

### 4.5. Significance of This Method

When the results of applying this method to the target workplace were compared with the results of applying other techniques, the proposed method was found to be less conservative, and CHARM was found to be less conservative than COSHH Essentials. It can be seen that CHARM and the proposed method exhibit a large difference in their results even though they both prioritize deriving the quantitative assessment results whenever possible.

To use the work environment monitoring results, exposure limit setting and work environment monitoring must be performed. To achieve both of these processes, toxicity assessments and a policy review process for the establishment of exposure standards are performed using experts, and many social resources are consumed; therefore, there are only a few substances that can be produced as a result of work environment monitoring. For a small number of substances for which quantitative risk assessment is possible, it is necessary to perform an accurate assessment using the corresponding values. This method determines the need for control by directly using the ratio between the exposure and the OEL. This approach has more intuitive and accurate characteristics than CHARM, which determines the necessity of control by banding the ratio between the exposure and the OEL and multiplying this band by the band of the OEL again.

## 5. Conclusions

In this study, we analyzed the conceptual validity of CHARM, a control banding tool developed by KOSHA to support workplace risk assessment, and developed a new comprehensive risk assessment method that complemented the logical errors of the concept and applied it to actual data. The optimized technique changed the method of quantitative assessment and evaluation criteria, matched the risk level with the required control level, and specified the use of control practice. The results were derived by classifying them according to the amount of information for each risk level, hazard band, and exposure band. The differences for each classification of the results were compared and analyzed, and a risk level or control approach for each similar technique was derived and compared.

The risk level calculated through control banding was more conservative than that calculated using the work environment monitoring results, and the result of assigning the Hazard statement was more conservative than that using the exposure standard band as a hazard band. In the case of the exposure band, the higher the band, the higher actual exposure amount did not appear, and it was confirmed that the estimated exposure for all bands was higher than the actual exposure. It was confirmed that the results for the target enterprises were conservative in the order of Control of Substances Hazardous to Health Essentials, CHARM, and this method.

In the case of risk levels or hazard bands, whose methods vary according to available information, fewer informational methods resulted in conservative results. There was a low risk of false negatives, resulting in the adoption of a lower-level control approach than necessary. However, since the control status of the enterprises could not be confirmed and the substances handled at the target enterprises were limited to highly toxic substances with a high regulatory level (which was a limitation of this study), a follow-up study should be performed to expand the data on target materials and supplemental information on the current control status of the enterprises.

## Figures and Tables

**Figure 1 ijerph-18-09395-f001:**
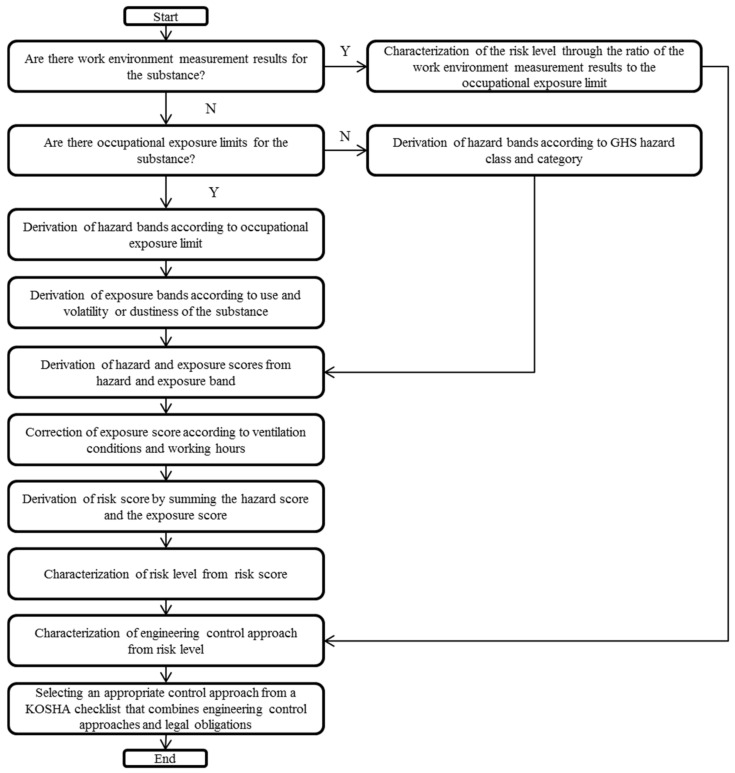
Flowchart of the risk assessment method.

**Figure 2 ijerph-18-09395-f002:**
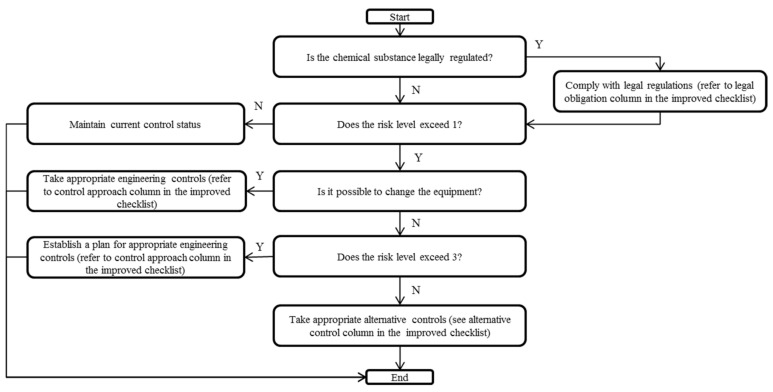
Flowchart of the process of selecting an appropriate control approach from the Korea Occupational Safety and Health Agency (KOSHA) checklist.

**Figure 3 ijerph-18-09395-f003:**
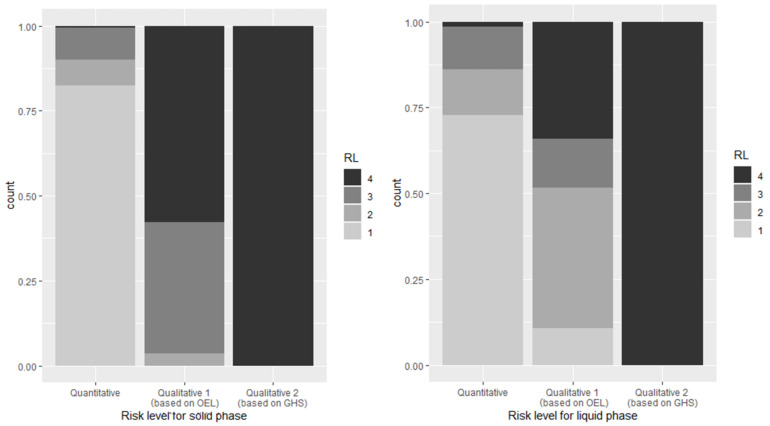
Risk level (RL) for each phase and branch based on available information.

**Figure 4 ijerph-18-09395-f004:**
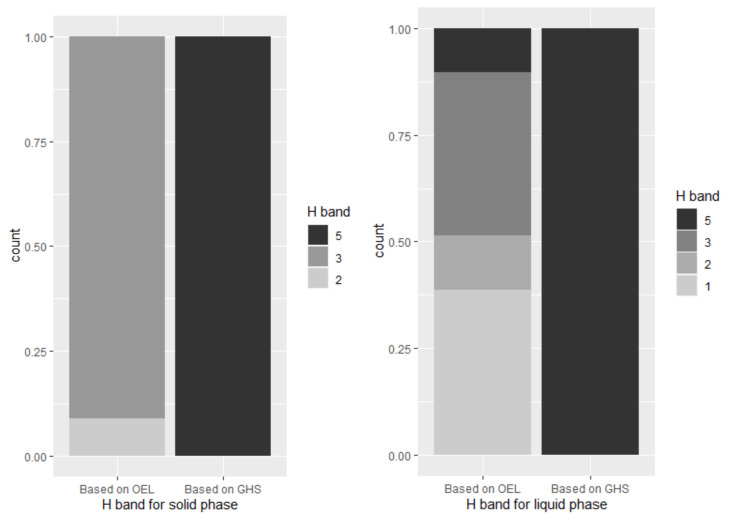
Hazard band (H band) for each phase and branch based on available information.

**Figure 5 ijerph-18-09395-f005:**
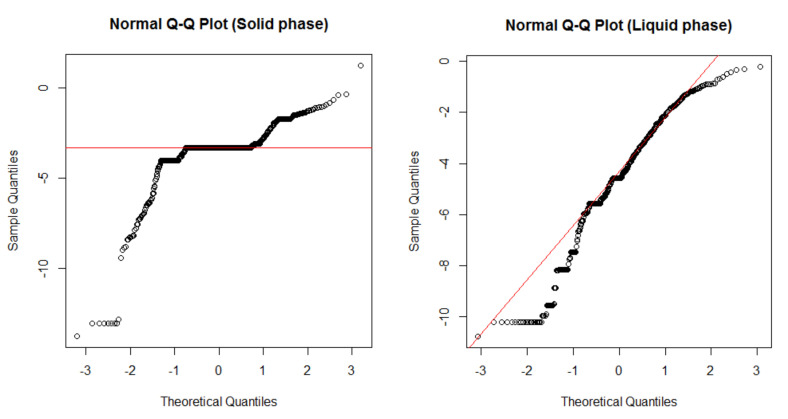
Normal Q–Q plot of exposure data for each phase.

**Figure 6 ijerph-18-09395-f006:**
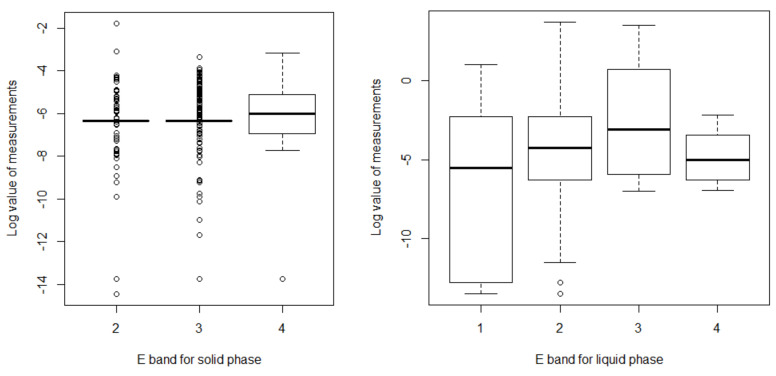
Exposure band (E band) for each phase.

**Figure 7 ijerph-18-09395-f007:**
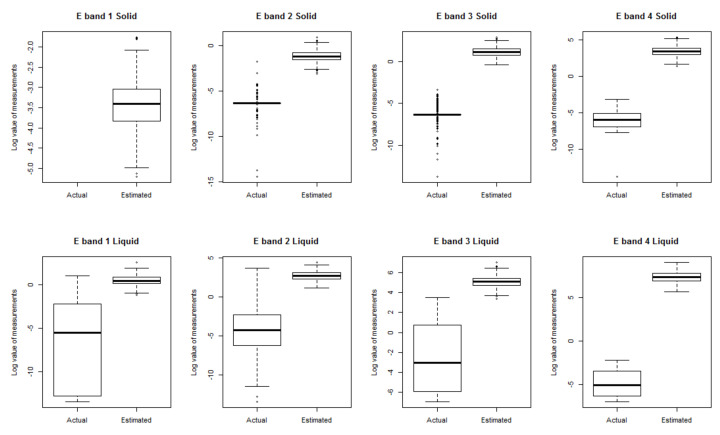
Comparison of actual and estimated exposure for each phase and band.

**Figure 8 ijerph-18-09395-f008:**
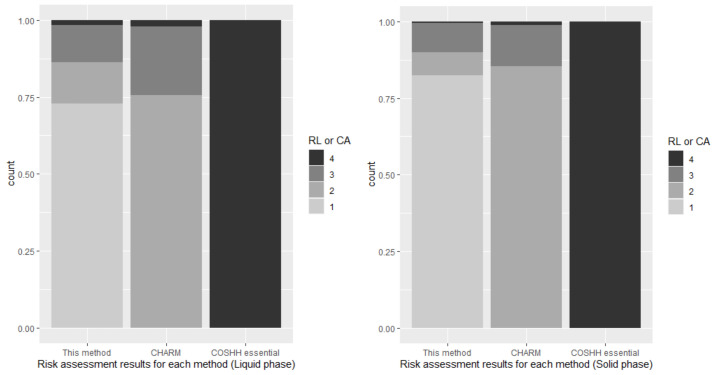
Risk assessment results for each phase and method.

**Table 1 ijerph-18-09395-t001:** Definition of risk levels derived from the ratio of the work environment monitoring results to the occupational exposure limit (OEL).

Ratio of the Mean of Work Environment Monitoring Results to Occupational Exposure Limit	Risk Level
0–0.05	1
0.05–0.15	2
0.15–0.5	3
>0.5 (or the ratio of the max work environment monitoring results to the OEL > 1)	4

**Table 2 ijerph-18-09395-t002:** Allocation of hazard statements to hazard bands and the associated target airborne concentration ranges represented by occupational exposure limits (OELs).

OEL/Target Airborne Concentration		Hazard Statements of Global Harmonized System of Classification and Labeling of Chemicals	Hazard Band
mg/m^3^	ppm		
<0.001	<0.05	H304 (May be fatal if swallowed and enters airways) H315 (Causes skin irritation) H319 (Causes serious eye irritation) H336 (May cause drowsiness or dizziness)	5
0.001–0.01	0.05–0.5	H302 (Harmful if swallowed) H312 (Harmful in contact with skin) H332 (Harmful if inhaled) H371 (May cause damage to organs)	4
0.01–0.1	0.5–5	H301 (Toxic if swallowed) H311 (Toxic in contact with skin) H314 (Causes severe skin burns and eye damage) H317 (May cause an allergic skin reaction) H318 (Causes serious eye damage) H331 (Toxic if inhaled) H335 (May cause respiratory irritation) H370 (Causes damage to organs) H373 (May cause damage to organs through prolonged or repeated exposure)	3
0.1–1	5–50	H300 (Fatal if swallowed) H310 (Fatal in contact with skin) H330 (Fatal if inhaled) H351 (Suspected of causing cancer) H360 (May damage fertility or the unborn child) H361 (Suspected of damaging fertility or the unborn child) H362 (May cause harm to breast-fed children) H372 (Causes damage to organs through prolonged or repeated exposure)	2
1–10 or more than that	50–500 or more than that	H334 (May cause allergy or asthma symptoms or breathing difficulties if inhaled) H340 (May cause genetic defects) H341 (Suspected of causing genetic defects) H350 (May cause cancer)	1

**Table 3 ijerph-18-09395-t003:** Definitions of exposure bands derived from amount and physical properties.

Phase	Amount and Physical Properties	Predicted Exposures	Exposure Band
Solid	Low/medium dustiness and gram use	0.01–0.1 mg/m^3^	1
	Low dustiness and kg/ton use or high dustiness and gram use	0.1–1 mg/m^3^	2
	Medium/high dustiness and kg use	1–10 mg/m^3^	3
	Medium/high dustiness and ton use	10–100 mg/m^3^ or more than that	4
Liquid	Low volatility and mL use	0.5–5 ppm	1
	Low volatility and L/m^3^ use or medium volatility and mL use	5–50 ppm	2
	Medium volatility and L/m^3^ use or high volatility and L use	50–500 ppm	3
	High volatility and m^3^ use	500–5000 ppm or more than that	4

**Table 4 ijerph-18-09395-t004:** Allocation of hazard and exposure bands to hazard and exposure scores.

Type	Band	Target or Predicted Airborne Concentration (Solid, mg/m^3^)	Log of the Max Value of Airborne Concentration Range	Score
Hazard	1	1–10	1	1
	2	0.1–1	0	2
	3	0.01–0.1	−1	3
	4	0.001–0.01	−2	4
	5	<0.001	−3	5
Exposure	1	0.01–0.1	−1	1
	2	0.1–1	−2	2
	3	1–10	1	3
	4	10–100	2	4

**Table 5 ijerph-18-09395-t005:** Correction of exposure scores according to ventilation condition and working hours.

Type	Condition	Correction Exposure Score
Ventilation	Poor ventilation	+1
	General ventilation	+0
	Local exhaust ventilation	−1
	Containment	−2
Working hours	Temporary work	−1
	Short-term work	−1
	Others	+0

**Table 6 ijerph-18-09395-t006:** Characterization of risk level from risk score.

Log Value of the Ratio of Predicted to Target Airborne Concentrations	Risk Score	Risk Level	General Description of Risk Level
<0	<4	1	Maintaining current control level
0	4	2	Applying engineering control 1 level higher
1	5	3	Applying engineering control 2 levels higher
>1	>5	4	Applying risk reduction measures more than 3 levels higher

**Table 7 ijerph-18-09395-t007:** Characterization of engineering control approach using risk levels and ventilation conditions.

Risk Level	Current Ventilation Condition			
	Poor ventilation	General ventilation	Local exhaust ventilation	Containment
1	General ventilation	General ventilation	Local exhaust ventilation	Containment
2	General ventilation	Local exhaust ventilation	Containment	Fundamental measures
3	Local exhaust ventilation	Containment	Fundamental measures	Fundamental measures
4	Containment	Fundamental measures	Fundamental measures	Fundamental measures

**Table 8 ijerph-18-09395-t008:** Classification of items in the control status checklist of Chemical Hazard Risk Management (CHARM) for working environments.

No.	Control Practice	Legal Obligation	Control Approach	Alternative Control
1	Can it be replaced with a substance that is less toxic (higher exposure limits) than the substance currently being handled?	-	Fundamental measures	-
2	If you are currently dealing with a carcinogenic substance, can it be replaced with a non-carcinogenic substance?	-	Fundamental measures	-
3	Is it possible to close the current hazardous substance handling process?	-	Fundamental measures	-
4	Can you reduce the amount of chemicals you use currently?	-	-	V
5	Is it possible to do wet work in the case of solid substances such as dust?	Compulsory for materials subject to permission	-	V
6	Is it possible to completely contain the hazardous substance handling process?	Compulsory for materials subject to permission	Containment	-
7	Is it possible to install a local exhaust ventilation system at the location of hazardous substances?	Compulsory for materials subject to permission/recommended for harmful substances requiring management	Local exhaust ventilation	-
8	Can the local exhaust system ventilation hood be installed in a booth type?	Compulsory for materials subject to permission	Local exhaust ventilation	-
9	Does the hood’s position protect the worker’s respiratory zone?	-	Local exhaust ventilation	-
10	Is it possible to equip a flange to increase collection efficiency?	-	Local exhaust ventilation	-
11	Does the control velocity of the local exhaust ventilation system meet the legal standards?	Compulsory for materials subject to permission/recommended for harmful substances requiring management	Local exhaust ventilation	-
12	Is the local exhaust ventilation system performance checked regularly?	Compulsory for materials requiring safety inspection	Local exhaust ventilation	-
13	Is it possible to equip a general ventilation (fan) system?	-	General ventilation	-
14	(If the process is affected by nearby processes) Can hazardous substance handling processes be operated in isolation from nearby processes and workplaces?	-	Fundamental measures	-
15	(If the process is affected by nearby processes) Is it possible to equip barriers to block air movement between hazardous material handling processes and nearby work sites?	-	Fundamental measures	-
16	Is it possible to change the process of the current hazardous substance handling tasks as automation or semi-automation?	-	Fundamental measures	-
17	Can the container for hazardous substances be stored in a separate storage location?	Recommended for harmful substances requiring management	-	-
18	Can hazardous substances be handled without direct contact?	-	Fundamental measures	-
19	Are health examinations conducted regularly?	Compulsory for materials subject to health examination	-	-
20	Are working environment measurements conducted regularly?	Compulsory for materials subjected to work environment monitoring	-	-
21	Are workers educated on handling chemicals?	Compulsory for all substances	-	-
22	Is personal respiratory protective equipment adequately provided?	Compulsory for materials subject to permission/recommended for harmful substances requiring management	-	V
23	Are workers wearing personal respiratory protective equipment during work?	-	-	V
24	Is the performance of personal respiratory protective equipment properly managed?	-	-	V
25	Have you installed signs to wear personal respiratory protective equipment in the workplace?	Compulsory for all substances	-	-
26	Are the protective equipment storage boxes installed and kept clean?	Compulsory for materials subject to permission/recommended for harmful substances requiring management	-	-
27	Are the chemical handling processes adequately clean?	Recommended for harmful substances requiring management	-	-
28	Have you kept and posted material safety data sheets for the chemicals you are handling?	Compulsory for all substances	-	-
29	Are warning signs attached to the handling chemical containers and packaging?	Compulsory for all substances	-	-

**Table 9 ijerph-18-09395-t009:** Harmful substances to be maintained at or below permissible levels as of 2014 in Korea.

Substances	Occupational Exposure Limit	Limit of Detection (KOSHA Guidance)	
Name	Unit		
Lead and inorganic compounds, as Pb	mg/m^3^	0.05	0.001733
Chromium (VI) compounds	mg/m^3^	0.05	0.0035
Toluene-2,4-diisocyanate (TDI)	ppm	0.005	5.61 × 10^−6^
Dimethylformamide	ppm	10	0.209074
Nickel (insoluble inorganic compounds, as Ni)	mg/m^3^	0.5	2.08 × 10^−6^
Trichloroethylene	ppm	50	0.003721
n-Hexane	ppm	50	0.028371
Formaldehyde	ppm	0.5	0.0038
Benzene	ppm	1	0.005217
Cadmium and compounds, as Cd	mg/m^3^	0.01	3.33 × 10^−5^

## Data Availability

The data presented in this study are available upon request from the corresponding author. The data are not publicly available due to privacy of monitored enterprises.

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
