# Peer review of "Review and Improvement of Chemical Hazard Risk Management of Korean Occupational Safety and Health Agency"

_ijerph, 2021, doi:10.3390/ijerph18179395_

Round 1

Reviewer 1 Report

Dear authors

Given the very high level of industrialization in South Korea, the relevance of your article is clear, so I will recommend to the Editor the publication of the manuscript.

There are however some improvements to be made to the manuscript. In the introduction, it is necessary to more accurately contextualize the information about Korean workers exposed to 45,000 chemical substances, according to the manuscript (line 32).

Specifically: 1) how many Korean workers are there in the industrial sector?; 2) how many workers in the industrial sector are exposed to chemicals that are hazardous to health? (for example, a guard who works in areas outside the factories will not be as exposed to these substances); 3) of the 45,000 chemical substances employed in South Korea's industries, how many (i.e. what percentage) are potentially harmful to health; 4) is there information about harmful substances most commonly used in industries (e.g. the 20 most common)?

If this information exists, please include it in the Introduction. This will give readers a more accurate dimension of the contamination levels that Korean workers are exposed to.

It was not clear to me what the 29 abbreviations in the "H statement" column of Table 2 mean (H304, H305 etc).

I've counted 18 acronyms throughout the manuscript: OEL, COSHH, KOSHA, CHARM, CB, H and E bands, ILO, HSE, IOHA, GHS, PEL, AIHA, INRS, BOHS/NVva, CEN, GDS, LEV, NIOSH. This tangle of acronyms and the definitions behind them make reading the text difficult and quite tiring, especially in the Introduction and Discussion. I strongly recommend that authors reduce the use of acronyms throughout the text, writing them out in full. For example, on page 16, the acronym COSHH (line 402) should be spelled out: Control of Substances Hazardous to Health, accompanied by the acronym (COSHH), which would allow readers to quickly understand its meaning when on the same page 16, this acronym is mentioned again, in lines 405, 410.412 and 421.

It was actually quite counterproductive to go back numerous times to the beginning of the manuscript to recall the meaning of the acronyms. Furthermore, I did not find the meaning of four acronyms in the text: HSE (page 3), GHS (page 4) NIOSH, GSD (page 15).

Author Response

Point 1: There are however some improvements to be made to the manuscript. In the introduction, it is necessary to more accurately contextualize the information about Korean workers exposed to 45,000 chemical substances, according to the manuscript (line 32).

Specifically: 1) how many Korean workers are there in the industrial sector?; 2) how many workers in the industrial sector are exposed to chemicals that are hazardous to health? (for example, a guard who works in areas outside the factories will not be as exposed to these substances); 3) of the 45,000 chemical substances employed in South Korea's industries, how many (i.e. what percentage) are potentially harmful to health; 4) is there information about harmful substances most commonly used in industries (e.g. the 20 most common)?

If this information exists, please include it in the Introduction. This will give readers a more accurate dimension of the contamination levels that Korean workers are exposed to. 

Response 1: Information on Korean workers was found and included in the introduction. Among the four specifically presented items, the total number of workers, the number of workers handling chemical substances, and the number of harmful factors managed by the government exist as public data. (line 35)

Point 2: It was not clear to me what the 29 abbreviations in the "H statement" column of Table 2 mean (H304, H305 etc).

Response 2: Hazard statement is a phrase that standardizes the hazard of a substance in the Global Harmonized System of classification and labeling of chemicals. It is often abbreviated as H statement and is coded in a specific form (Hxxx).
The full name of the concept is used as the column heading in Table 2 and an explanation of what each H statement indicates in the table is added. (line 161-Table2)

Point 3: I've counted 18 acronyms throughout the manuscript: OEL, COSHH, KOSHA, CHARM, CB, H and E bands, ILO, HSE, IOHA, GHS, PEL, AIHA, INRS, BOHS/NVva, CEN, GDS, LEV, NIOSH. This tangle of acronyms and the definitions behind them make reading the text difficult and quite tiring, especially in the Introduction and Discussion. I strongly recommend that authors reduce the use of acronyms throughout the text, writing them out in full. For example, on page 16, the acronym COSHH (line 402) should be spelled out: Control of Substances Hazardous to Health, accompanied by the acronym (COSHH), which would allow readers to quickly understand its meaning when on the same page 16, this acronym is mentioned again, in lines 405, 410.412 and 421.

It was actually quite counterproductive to go back numerous times to the beginning of the manuscript to recall the meaning of the acronyms. Furthermore, I did not find the meaning of four acronyms in the text: HSE (page 3), GHS (page 4) NIOSH, GSD (page 15).

Response 3: In this paper, we will avoid using acronyms for words with a small frequency. (CB, HSE, IOHA, PEL, INRS, BOHS/NVva, CEN, LEV, NIOSH)
When an acronym is commonly used in occupational health or is part of a proper noun, it is difficult not to use the acronym at all.
However, if it is used multiple times on one page, such as COSHH in your example, the full name is used at the very beginning of the page to improve readability. (OEL, COSHH, CHARM, H and E band)
Acronyms whose definitions were missing were described by adding definitions (GHS, GSD) or by unraveling the definitions (HSE, NIOSH).

Reviewer 2 Report

In this study, the authors analyzed the conceptual validity of CHARM, a CB tool developed by KOSHA to support workplace risk assessment, and developed a new comprehensive risk assessment method that complemented the logical errors of the concept and applied it to actual data. But I do not think they are interesting to readers. And I believe it is much like a manual and use report only. The author should find new innovations for readers.

Author Response

Korea is a highly industrialized country, and I think that the current state of chemical management in Korea will attract the attention of many researchers. Through this revision, some information about the current situation has been added.
Although this evaluation technique is composed of simple evaluation criteria, it may look like a manual, but simplicity is an essential element in terms of the technique used in the workplace. have. Through this revision, the differences from existing techniques were specifically specified in the abstract, results, and conclusions to help understand the innovativeness of the paper.

Reviewer 3 Report

It is an interesting topic. However, I have several questions. I will detail my comments below.

Frist, the manuscript aims to improve the CHARM. However, the improvement of it did not clearly showed in this manuscript. Especially not mentioned in the ‘Abstract’ and the ‘Conclusions’ sections.

Secondly, the developed method seems to have lots limits that make the significance and innovation of this study are less. The difference between the optimized method and the original CHARM needed to be explained more specifically.

Last, the ‘Discussion’ part is too long. It is suggested to add a subtitle to increase readability.

Author Response

Point 1: Frist, the manuscript aims to improve the CHARM. However, the improvement of it did not clearly showed in this manuscript. Especially not mentioned in the ‘Abstract’ and the ‘Conclusions’ sections. 

Response 1: In connection with the second point, specific improvements were added to the text, and the 'Abstract' and 'Conclusion' sections were also revised. (line 8, 245 and 547)

Point 2: Secondly, the developed method seems to have lots limits that make the significance and innovation of this study are less. The difference between the optimized method and the original CHARM needed to be explained more specifically.

Response 2: A detailed explanation of the difference between the optimized method and the original CHARM has been added to the text. (line 245)

Point 3: Last, the ‘Discussion’ part is too long. It is suggested to add a subtitle to increase readability.

Response 3: We added a subtitle to the discusstion part. (line 355, 388, 441, 476, and 524)

Round 2

Reviewer 2 Report

Within this version, the author try to give a nice method and idea for chemical hazard risk management of Korean occupational safety and health agency. The results and abstract are rewirte and sound much valuable. No further comment except grammar mistake checking.

Reviewer 3 Report

ok. I have no problem.